# Spatiotemporal Distribution of Total Suspended Matter Concentration in Changdang Lake Based on In Situ Hyperspectral Data and Sentinel-2 Images

**Zuoyan Gao** [1,2,3], **Qian Shen** [2,4,*], **Xuelei Wang** [5], **Hongchun Peng** [1], **Yue Yao** [2], **Mingxiu Wang** [2], **Libing Wang** [2], **Ru Wang** [1,2], **Jiarui Shi** [1,2], **Dawei Shi** [3] and **Wenguang Liang** [6]

1   School of Marine Technology and Geomatics, Jiangsu Ocean University, Lianyungang 222005, China; 2019220207@jou.edu.cn (Z.G.); penghc@lzb.ac.cn (H.P.); 2018224043@jou.edu.cn (R.W.); 2018224033@jou.edu.cn (J.S.)
2   Key Laboratory of Digital Earth Science, Aerospace Information Research Institute, Chinese Academy of Sciences, Beijing 100094, China; yaoyue@aircas.ac.cn (Y.Y.); Mingxiuwang@ahnu.edu.cn (M.W.); 2019212381@nwnu.edu.cn (L.W.)
3   Lianyungang Meteorological Bureau, Lianyungang 222006, China; lidl@aircas.ac.cn
4   College of Resources and Environment, University of Chinese Academy of Sciences, Beijing 100049, China
5   Center for Satellite Application on Ecology and Environment, Ministry of Ecology and Environment (MEE), Beijing 100094, China; wxuelei@bnu.edu.cn
6   Lake Research Institute, Jiangsu Water Conservancy Research Institute, Nanjing 210017, China; zhangyuting211@mails.ucas.ac.cn
*   Correspondence: shenqian@radi.ac.cn

**Abstract:** The concentration of total suspended matter (TSM) is an important parameter for evaluating lake water quality. We determined in situ hyperspectral data and TSM concentration data for Changdang Lake, China, to establish a TSM concentration inversion model. The model was applied using 60 Sentinel-2 images acquired from 2016 to 2021 to determine the temporal and spatial distribution of TSM concentration. Remote sensing images were also utilized to monitor the effect of ecological dredging in Changdang Lake. The following results were obtained: (1) Compared with four existing models, the TSM concentration inversion model established in this study exhibited higher accuracy and was suitable for Changdang Lake. (2) TSM concentrations obtained for the period 2016–2021 were higher in spring and summer, and lower in autumn and winter. (3) The dredging process influenced a small area of the surrounding water body, resulting in higher TSM concentrations. However, a subsequent reduction in TSM concentrations indicated that the ecological dredging project might improve the water quality of Changdang Lake to a considerable extent. Therefore, it was inferred that the use of Sentinel-2 images was effective for the long-term monitoring of water quality changes in small and medium-sized lakes.

**Keywords:** Changdang Lake; total suspended matter concentration; Sentinel-2 image; ecological dredging

## 1. Introduction

Total suspended matter (TSM) refers to solid matter suspended in water, and predominantly comprises insoluble organic matter and sediment. The concentration of TSM is an important index utilized for evaluating the water quality of a lake, as it affects the optical properties of water through absorption and scattering [1]. As one of the three main components considered in the remote sensing-based analysis of water quality, TSM is deemed as the main cause of water turbidity [2–4]. Therefore, real-time and accurate monitoring of TSM concentration is crucial for determining lake water quality in a timely manner, for exploring the factors driving changes in TSM concentration, and for formulating more effective water quality improvement strategies.

Traditional water quality monitoring is conducted through on-site sampling and subsequent laboratory analysis. Although this method is highly accurate, it is also time-consuming and expensive and cannot be used to determine the overall spatial distribution of water quality [5,6]. Several studies have reported the utilization of high and low spatial resolution satellite remote sensing images to retrieve data on TSM concentrations over wide sea areas and inland water bodies, for example, moderate resolution imaging spectrometer (MODIS) images, medium resolution imaging spectrometer (MERIS) images, HJ1A/SHI data, Satellite Pour l'Observation de la Terre (SPOT) series data, visible infrared imaging radiometer (VIIRS) data, Landsat series data, and Sentinel-2 multispectral instrument (MSI) data [6–17]. Affected by its spatial resolution, different satellite images can detect different minimum widths of rivers and lakes. Generally, 3~5 pixels is the minimum width of water bodies that can be adequately analyzed from satellite images with fewer adjacency effects. Images from Sentinel-2 are more detailed than those from Landsat and MODIS sensors. For example, for rivers or lakes with a width of 1000 m, only 4 pixels can be observed in MODIS images with a spatial resolution of 250 m, 33 pixels can be observed in Landsat OLI images with a spatial resolution of 30 m, but 100 pixels can be observed in Sentinel-2 images with a spatial resolution of 10 m. Therefore, Sentinel-2 images can be used to monitor the water bodies in medium and small lakes more accurately, without the influence of adjacency effects; however, there have been very few studies on using Sentinel-2 images to monitor TSM concentration in medium and small lakes. As MODIS is low spatial resolution data with a short revisit period, it is generally utilized for conducting quantitative monitoring and long-term series analysis of TSM concentrations in large lakes [7,12,18]. The Sentinel-2 series of satellites, which were first launched in June 2015, comprises two satellites, namely Sentinel 2A and Sentinel 2B, each carrying a multispectral imager. They have a revisit period of five days and contain 13 spectral bands, with a width of 290 km and a spatial resolution of 10 m. Owing to its temporal and spatial resolution, the Sentinel-2 satellite can be utilized to support research for determination of long-term TSM concentration variations in small lakes and reservoirs [19].

At present, four major models exist for estimation of the TSM concentration of inland water bodies on the basis of remote sensing: empirical, semi-empirical, semi-analytical, and artificial intelligence models [20–23]. The empirical model is used by inverting the TSM concentration based on on-site measured data and remote sensing data to establish a relevant mathematical statistical relationship. This method is relatively simple and widely used, but its applicability is undoubtedly limited by time and location [24,25]. The semi-empirical model is utilized to analyze the spectral characteristics of water elements to establish a statistical relationship between the characteristic bands or band combination of remote sensing data and synchronously measured TSM concentrations, thus aiding relatively more reliable inversion results [26]. The semi-analytical model is based on the mechanism of radiative transfer in water and relies on highly accurate atmospheric correction of remote sensing data [21,22,27]. Finally, the artificial intelligence model is a TSM concentration inversion model based on neural networks, genetic algorithms, and support vector machines. However, the model results are markedly affected by the training samples. It requires a substantial number of training samples to ensure model accuracy [28–30]. In this study, a semi-empirical method was utilized to establish an inversion model of the TSM concentration in Changdang Lake in China.

A semi-empirical method has been widely utilized for the inversion of TSM concentrations in large sea areas and inland waters. For example, Miller and McKee [7] established a linear estimation model based on the TSM concentration in the northern Gulf of Mexico by analyzing MODIS Terra band 1 data (620–670 nm). Further, this model could be utilized to examine the migration of small water bodies in the Gulf and estuary. Shi et al. [12] studied the TSM concentration in Taihu Lake, China, and found a significant correlation between MODIS Aqua reflectance data at 645 nm and the measured TSM concentration. Several researchers have utilized remote sensing reflectance spectra in the characteristic band of 700–850 nm to determine TSM concentrations [31–33]. For example, Zheng et al. [34] re-

ported the development of three estimation models of TSM concentration based on a strong correlation between the near-infrared (775–900 nm) reflectance of Landsat OLI, ETM+/TM, and MSS data and the measured TSM concentration. Moreover, Sun et al. [35] reported the highest correlation between measured reflectance and TSM concentration at 725 nm in Taihu Lake. Li and Wan [36] also observed a good correlation between the near-infrared band and TSM concentration in Chaohu Lake. Significant correlations have been observed between other bands or band combinations of remote sensing images and measured TSM concentrations [10,19,37,38]. However, few studies have reported the use of Sentinel-2 images to quantitatively estimate TSM concentrations in medium and small lakes, which limits their applications in water quality monitoring. Changdang Lake covers an area of 85 square kilometers. Using the Sentinel-2 images with a spatial resolution of 10 m to monitor Changdang Lake, we could extract a sufficient number of pure water pixels to study the lake, without the influence of the adjacency effect. However, if we use the MODIS image with a spatial resolution of 1000 m to monitor Changdang Lake, the area of the lake can be greatly reduced and the research results will be affected by the edge adjacency effect.

In this study, satellite remote sensing images acquired for the period 2016–2021 have been utilized to explore the temporal and spatial distribution of TSM concentrations in Changdang Lake; the lake has been characterized as a small to medium lake and has been reported to be affected by both natural environmental factors and anthropogenic factors. The effect of ecological dredging of the lake was monitored via remote sensing over the same period, and the factors driving seasonal variations in TSM concentration were analyzed. The purpose of this study was to (1) establish a quantitative remote sensing inversion model of TSM concentration in Changdang Lake based on Sentinel-2 data; (2) generate a spatiotemporal distribution map of TSM concentrations in relation to ecological dredging of the lake based on Sentinel-2 data acquired from 2016 to 2021; and (3) analyze the seasonal variations in and driving factors affecting TSM concentration.

## 2. Data and Methods

### 2.1. Study Area

Changdang Lake, also known as Tao Lake, is located in the southeast of Jintan District, Jiangsu Province, China, in the upper reaches of the Taihu Lake basin, at $119°30'0''–119°40'0''$ E longitude and $31°30'0''–31°40'0''$ N latitude. Since the 1980s, rapid economic development has led to the exertion of effects by industrial wastewater, domestic sewage, farmland drainage, and purse seine farming on Changdang Lake, resulting in the accumulation of sediments containing harmful substances and pollution of the lake water [39,40]. To improve the water environment and to maintain the ecological health of Changdang Lake, the local government has implemented the principle of ecological dredging to remove harmful sediment from its bottom. Changdang Lake covers an area of 85 km$^2$, presenting an annual average water level of 1.2–1.5 m. As a typical shallow lake, it is considered vital in various scenarios such as flood discharge, drinking water supply, tourism, and fishery production. Several rivers enter and leave Changdang Lake, including the Fangluo, Xinhe, Dapu, Baishi, Renhe, and Zhuangyang Rivers from the northwest to the southwest shores. In addition, the Huangli, Beigan, and Youshan Xinhe rivers enter from the northeast to southeast shores.

### 2.2. Data Sources

#### 2.2.1. Measured Data

In this study, a field experiment was conducted on the surface of Changdang Lake on 3 May 2020. It included measurement of in situ hyperspectral data, water sampling, and on-site water quality measurements, with a total inclusion of 20 sampling points (Figure 1). During the experiment, the wind speed (measured using an anemometer in m/s) and transparency (measured as the Secchi depth in cm) were measured in situ. The measured transparency of Changdang Lake ranged from 9.7 cm to 31 cm, following which data on the point coordinates were recorded and scene photographs were acquired. At each point,

water samples were collected and stored in cold storage, following which they were further brought back to the laboratory for total TSM concentration measurement by considering the calcined filter membrane weighing method [41].

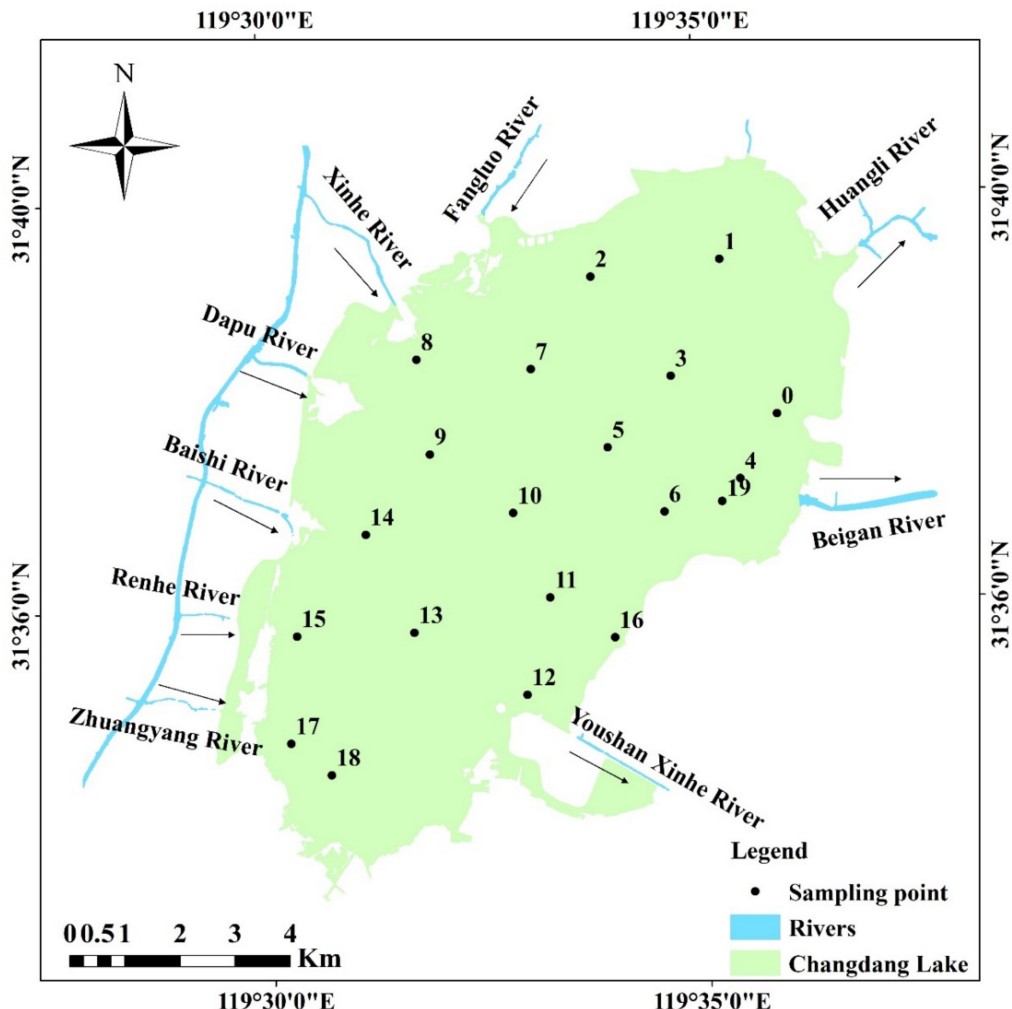

**Figure 1.** Location of the study area and sampling points.

To measure the surface reflectance spectra, the FieldSpec 4 Hi-Res surface spectrometer produced by Analytical Spectral Devices (ASD) was implemented [42,43]. This device could be continuously used to measure spectra in the wavelength range of 350–2500 nm while avoiding the effects of direct sunlight and reflection. In this case, the surface reflectance spectra of each measurement sampling point were maintained when the measurement azimuth was 45° or 135°. Equation (1) was utilized to eliminate the effects of the specular reflection of light on the water surface from the measured spectra and to calculate the reflectance of the sampling point:

$$R_{rs}(\lambda) = \frac{L_t(\lambda) - r \times L_{sky}(\lambda)}{L_p(\lambda)/\rho_p \times \pi} \tag{1}$$

where $L_t(\lambda)$ represents the upward radiance brightness of the water body; $r$ indicates the sky reflectance, which is determined by the sun's position, observation geometry, wind speed, wind direction, and other factors, and has been calculated according to the Fresnel formula. $L_{sky}$ represents the downward radiance brightness of the sky, which is directly measured using ASD spectrometry during the field experiment. $L_p(\lambda)$ indicates the reference board radiation brightness, and $\rho_p$ denotes the standard reference board

reflectance obtained via laboratory-based calibration. The resulting spectra of measured water reflectance have been subsequently presented as a series of curves.

2.2.2. Remote Sensing and Meteorological Data

Changdang Lake occupies a small area; therefore, considering the monitoring range of remote sensing data, Sentinel-2 images with a spatial resolution of 10 m were used in this study. Sentinel-2 data of Changdang Lake acquired from 2016 to 2021 were obtained from the European Space Agency (ESA) data distribution system (https://scihub.copernicus.eu/dhus/#/home, accessed on 3 March 2021) and the United States Geological Survey (USGS) website (https://earthexplorer.usgs.gov/, accessed on 12 November 2020). The L2A surface reflectance product information reported from 2019 to 2021 was obtained from the ESA. Images with 10 m spatial resolution were then resampled using the SNAP software, following which the image band was synthesized and the water body information was extracted. After water body information extraction, the normalized remote sensing-based reflectance of the Sentinel-2 image was estimated by dividing the extracted water body by 10,000, using band math. As the images pertaining to the period 2016–2018 were not available from ESA, the L1C top of atmosphere (TOA) reflectance product was obtained from the USGS website. For L1C products, the independent atmospheric correction module Sen2cor developed by ESA was used to correct the values for the atmosphere, and the L2A surface reflectance products were obtained. The remote sensing reflectance of the images was obtained by repeating the preprocessing steps described above. A total of 60 images were obtained by quality inspection of the data, which included removal of images exhibiting effects exerted by clouds, flares, and other factors.

The final 60 Sentinel-2 images were classified according to spring (images acquired from March to May), summer (images acquired from June to August), autumn (images acquired from September to November), and winter (images acquired from December of the previous year and January and February of the following year). Owing to the overcast and rainy weather in the region of Changdang Lake, and due to the influence of algal blooms and solar flare, a smaller number of Sentinel-2 data were available for a few seasons (especially in summer). The number of Sentinel-2 images for each season acquired from 2016 to 2021 is shown in Table 1.

**Table 1.** Sentinel-2 data obtained in the four seasons from 2016 to 2021.

| Year | Image Quantity | | | |
|------|--------|--------|--------|--------|
| | **Winter** | **Spring** | **Summer** | **Autumn** |
| 2016 | 1 | - | - | 1 |
| 2017 | 2 | 2 | 1 | 2 |
| 2018 | 5 | 5 | - | 3 |
| 2019 | 3 | 2 | 5 | 5 |
| 2020 | 6 | 3 | 2 | 4 |
| 2021 | 6 | 2 | - | - |

Daily mean wind speed and daily precipitation data acquired from 2017 to 2019 were obtained from the Liyang monitoring station (119°30′0″ E, 31°25′0″ N, station number: 58345), which was the nearest meteorological station to Changdang Lake. The data were downloaded from the National Greenhouse Data System (http://data.sheshiyuanyi.com/WeatherData/, accessed on 23 December 2020), a Chinese website.

*2.3. Research Methods*

In this study, a relatively stable TSM concentration inversion model was established by obtaining the in situ hyperspectral data and Sentinel-2 image data of Changdang Lake, and the model was applied to Sentinel-2 images from 2016 to 2021 to realize the long-term

monitoring of TSM concentration in Changdang Lake. The technical flow chart of this study is shown in Figure 2.

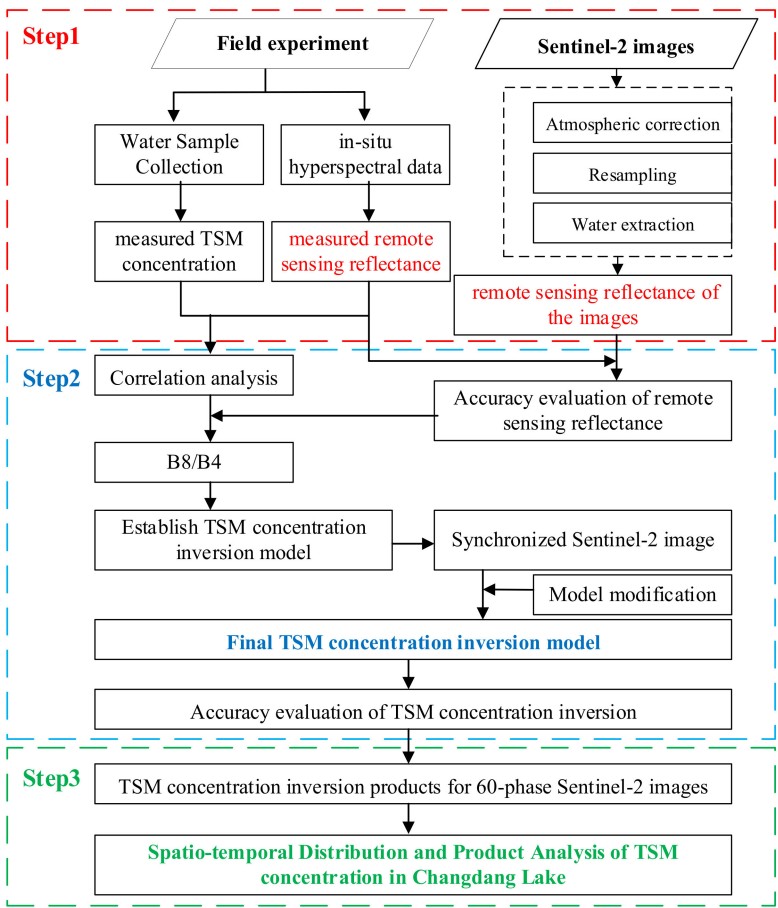

**Figure 2.** The technical flow chart of this paper.

### 2.3.1. Sentinel-2 Image Recorrection

Owing to the strong absorption of water observed in the near-infrared band and short-wave infrared band, the remote sensing reflectance in the short-wave infrared band can be ignored for turbid water, whereas the remote sensing reflectance in the near-infrared band can be ignored for clean water. Shi and Wang [44,45] conducted research based on this theory. As the water body of Changdang Lake was relatively turbid, this study utilized the assumption of negligible remote sensing reflectance in the short-wave infrared band to perform recorrection of the Sentinel-2 images.

The method used for correcting inland water information proposed by Shenglei et al. [46] was implemented to correct the uncertainty caused by insufficient atmospheric correction in Sentinel-2 images. Specifically, the value of the recorrected remote sensing reflectance was obtained by subtracting the minimum positive reflectance of the short-wave infrared band from each pixel of Sentinel-2 data and dividing by $\pi$. Equation (2) has been expressed as follows:

$$R_{rs}^r(\lambda) = \frac{R(\lambda) - \min(R_{SWIR})}{\pi} \qquad (2)$$

where $R_{rs}^r(\lambda)$ represents the recorrected remote sensing reflectance corresponding to the wavelength ($\lambda$) band of the Sentinel-2 data center, $R(\lambda)$ denotes the surface reflectance of the band with a center wavelength of $\lambda$, and $\min(R_{SWIR})$ indicates the minimum normal reflectance of the short-wave infrared band.

2.3.2. Accuracy Evaluation Index

The remote sensing reflectance data of the recorrected Sentinel-2 images were compared with the measured remote sensing reflectance data, and the results were evaluated according to the average relative error (MRE) and root mean square error (RMSE). The two accuracy evaluation indices were also utilized to evaluate the accuracy of the Changdang Lake TSM concentration inversion model. The equations used for calculation have been expressed as follows:

$$MRE = \frac{1}{n}\sum_{i=1}^{n}\frac{|X_i - X_j|}{X_j} * 100\% \tag{3}$$

$$RMSE = \sqrt{\frac{\sum_{i=1}^{n}(X_i - X_j)^2}{n}} \tag{4}$$

where $X_i$ represents the image value, $X_j$ represents the measured value, and $n$ represents the number of sampling points.

2.3.3. Construction of the TSM Concentration Inversion Model

(1)  Remote sensing reflectance simulation

The spectral data measured in the field exhibited the generation of a continuous curve with an interval of 1 nm, whereas Sentinel-2 satellite data exhibited a discrete multispectral band. Generally, after obtaining information on the remote sensing reflectance of the water body through indoor processing with reference to the spectral response function of each band of the Sentinel-2 image, the measured remote sensing reflectance was found to be equivalent to the multispectral band of the Sentinel-2 image. Furthermore, the equivalent reflectance was obtained, which was considered convenient for subsequent data processing. Equation (5) has been calculated as follows:

$$R_{rs_{eq}} = \frac{\int R_{rs}(\lambda) f_{SRE}(\lambda) F_0(\lambda) d(\lambda)}{\int f_{SRE}(\lambda) F_0(\lambda) d(\lambda)} \tag{5}$$

where $R_{rs_{eq}}$ represents the equivalent reflectance of the satellite band, $R_{rs}(\lambda)$ represents the measured remote sensing reflectance, $f_{SRE}(\lambda)$ denotes the spectral response function of the satellite, and $F_0(\lambda)$ denotes the solar irradiance outside the atmosphere. According to the band setting of the synchronous transit Sentinel-2A MSI image in the field test, the measured spectra were found to be equivalent to the remote sensing reflectance of nine bands. The measured reflectance spectra and Sentinel-2 equivalent reflectance spectra are displayed in Figure 3.

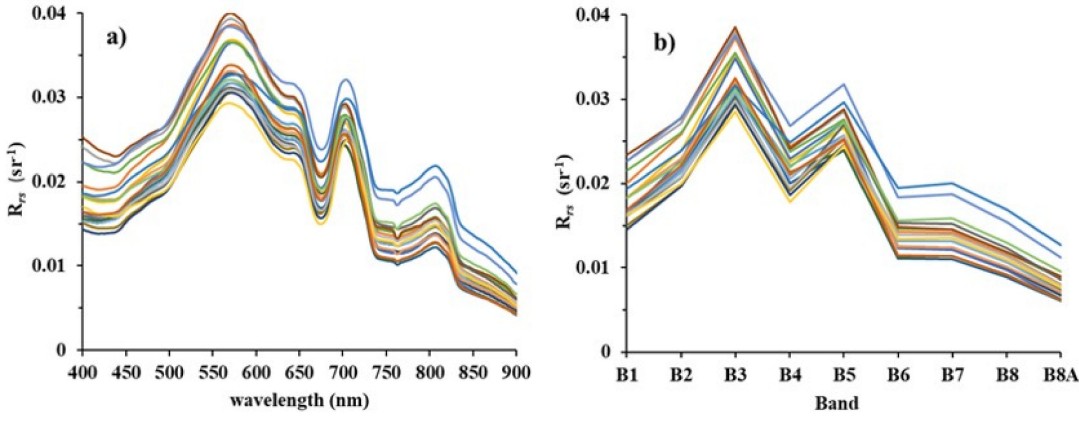

**Figure 3.** (**a**) Measured reflectance spectra of surface water, and (**b**) the equivalent reflectance spectra of Sentinel-2 in Changdang Lake reported on 3 May 2020.

(2) TSM concentration inversion model of Changdang Lake

Ground-measured remote sensing reflectance data were utilized to fit the multispectral data of Sentinel-2 images and to construct the TSM concentration inversion model. Various algorithms have been used to establish TSM concentration inversion models, including the single band (red band or near-infrared band) or band ratio (near-infrared band/red band or near-infrared band/green band) method [9,10,12,34].

In this study, by analyzing the correlation between the measured reflectance of each band or band combination and the measured value of TSM concentration, we inferred that bands B8 (842 nm) and band combinations B8/B4 (842 nm/665 nm) exhibited strong correlations with TSM concentration. In general, a single band could easily be affected by the atmosphere, leading to unstable inversion results being obtained. In this study, the use of the band ratio (B8/B4) method effectively reduced the impact of the atmosphere. A total of 20 sets of data were obtained from the field experiment of Changdang Lake on 3 May 2020. In this study, two methods were utilized to construct the TSM concentration inversion model and verify its accuracy. Finally, the optimal model was selected for the inversion of TSM concentration.

First, 2/3 of the sample points were randomly selected to establish the model with the measured TSM concentration, and the remaining 1/3 was used to verify the accuracy of the model. This method may be affected by sampling deviation. In order to verify the stability of the first modeling method, the second TSM concentration modeling method was tried: the k-fold cross-validation approach [47]. Here, we chose k = 5; that is, 20 data were randomly divided into 5 groups, each set having 4 data points. Further, one of them was selected as the verification dataset and the remaining 4 groups as the modeling set. Five models of TSM concentration were obtained by repeating the above process 5 times, and the best model was selected as the final model.

(a) 2/3 sample modeling

Fourteen points were randomly selected to establish the TSM concentration inversion model, as shown in Figure 4a. The remaining 6 verification samples were utilized to verify the model, and the evaluation results are shown in Figure 4b; MRE = 24.56%, RMSE = 18.72 mg/L.

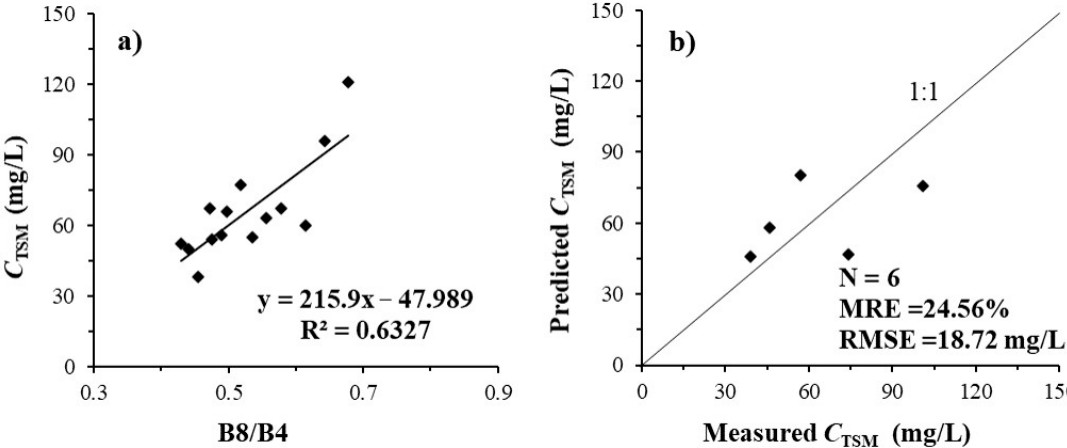

**Figure 4.** (**a**) The TSM concentration inversion model established by method 1, and (**b**) the accuracy evaluation of the model by verification samples.

(b) Five-fold cross-validation approach

Using the five-fold cross-validation method, five TSM concentration models were established. Further, the verification samples were utilized to evaluate the accuracy of the model. The model with the best accuracy was selected, as displayed in Figure 5a. The accuracy evaluation results of the verification samples to the model were as follows:

Figure 5b, MRE = 15.14%, RMSE = 10.07 mg/L. The results exhibited that the accuracy of the TSM concentration inversion model of Changdang Lake obtained by using five-fold cross-validation method is better.

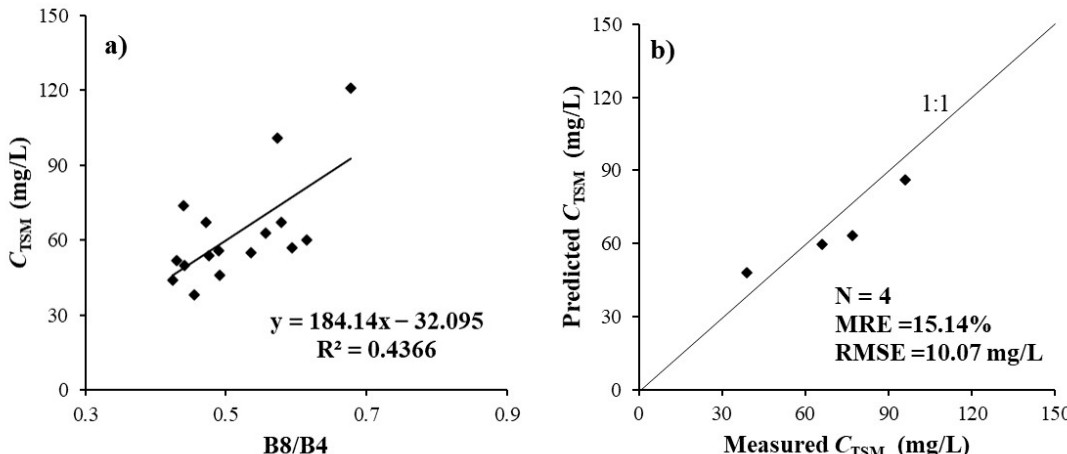

**Figure 5.** (**a**) The TSM concentration inversion model established by method 2, and (**b**) the accuracy evaluation of the model by verification samples.

(c)   Modification of TSM concentration inversion model based on synchronous Sentinel-2 Image

As few differences exist between the measured remote sensing reflectance and the reflectance of the Sentinel-2 images, the model may be overestimated when it is directly applied to Sentinel-2 images. In order to reduce the error of inversion, we modified the TSM concentration inversion model based on synchronous Sentinel-2 image reflectance, and the modified model expression is Equation (6). This model was used as the final model to retrieve the long-term TSM concentration in Changdang Lake.

$$C_{TSM} = 184.14 * \left( \frac{B8}{B4} \right) - 59.716 \tag{6}$$

where $C_{TSM}$ represents the retrieved TSM concentration (mg/L). The model input is the remote sensing reflectance ($R_{rs}$, unit is $sr^{-1}$) image obtained after atmospheric recorrection. B4 (665 nm) is indicated by the red band, and B8 (842 nm) is indicated by the near-infrared band.

## 3. Results

### 3.1. Accuracy Evaluation

#### 3.1.1. Evaluation of Sentinel-2 Image Recorrection Results

According to Equation (2), the Sentinel-2 image was recorrected in this study. In order to evaluate the effectiveness of the Sentinel-2 image recorrection method, according to the longitude and latitude of the sampling points in the field experiment, we extracted the remote sensing reflectance of each band from the Sentinel-2 image and compared the measured remote sensing reflectance with the Sentinel-2 image reflectance before and after recorrection. MRE and RMSE of Sentinel-2 images before and after recorrection could be obtained. The correction results are presented in Table 2. The error between the measured reflectance and image reflectance was markedly reduced after performing recorrection of the Sentinel-2 image. The MRE and RMSE of the B8/B4 band after recorrection were 33.51% and 0.17, respectively. The recorrection method of the Sentinel-2 image could improve the accuracy of TSM concentration inversion in Changdang Lake to some extent.

**Table 2.** Comparison of Sentinel-2 data errors before and after recorrection.

| Band/Band Combination | Before and after Image Recorrection | MRE (%) | RMSE (sr$^{-1}$) |
|---|---|---|---|
| B4 (665 nm) | Before recorrection | 25.57 | 0.0056 |
| | After recorrection | 8.51 | 0.0020 |
| B8 (842 nm) | Before recorrection | 80.31 | 0.0088 |
| | After recorrection | 40.81 | 0.0045 |
| B8/B4 | Before recorrection | 43.67 | 0.2200 |
| | After recorrection | 33.51 | 0.1700 |

3.1.2. Accuracy of Inversion Results Based on Synchronized Transit Images

The TSM concentration inversion model of Changdang Lake was established according to Equation (6). The model was applied to the B8/B4 band combination of the Sentinel-2 image on 3 May 2020. Furthermore, the predicted value of Sentinel-2 image inversion was obtained. The accuracy of the inversion model was evaluated by analyzing the measured and predicted values of TSM concentration (Figure 6). When the TSM concentration inversion model established in this study was used in conjunction with Sentinel-2 images, the difference between predicted and measured TSM concentration was found to be small. The MRE and RMSE values obtained were 20.49% and 14.96 mg/L, respectively, indicating good accuracy.

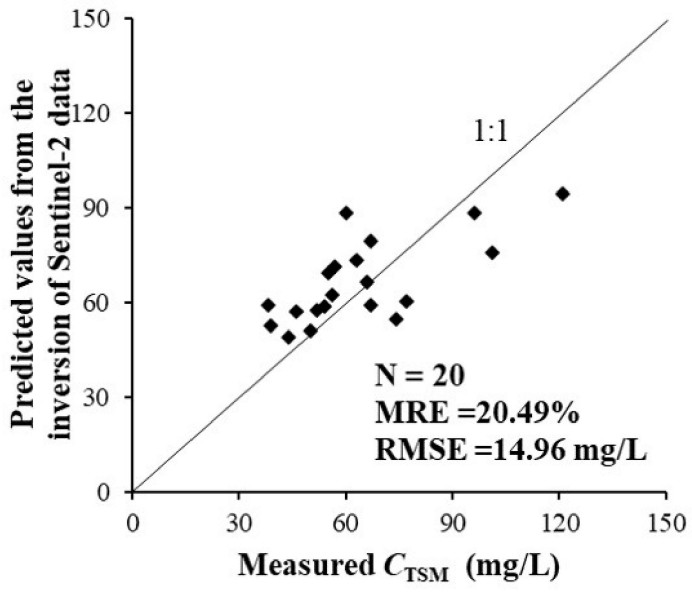

**Figure 6.** Comparison between the measured TSM concentration and predicted values based on the inversion of Sentinel-2 data.

*3.2. Temporal and Spatial Distribution of TSM Concentration*

As a typical shallow lake, Changdang Lake is susceptible to wind speed, precipitation, and human factors. In addition, the effects subsequently lead to the occurrence of substantial changes in the spatial distribution of TSM. After successful image verification, the TSM concentration inversion model of Changdang Lake was considered using the 60 Sentinel-2 images acquired from 2016 to 2021. The resulting distribution of TSM concentrations in Changdang Lake exhibited substantial spatial variations, ranging from 0.1 to 200 mg/L (Figure 7).

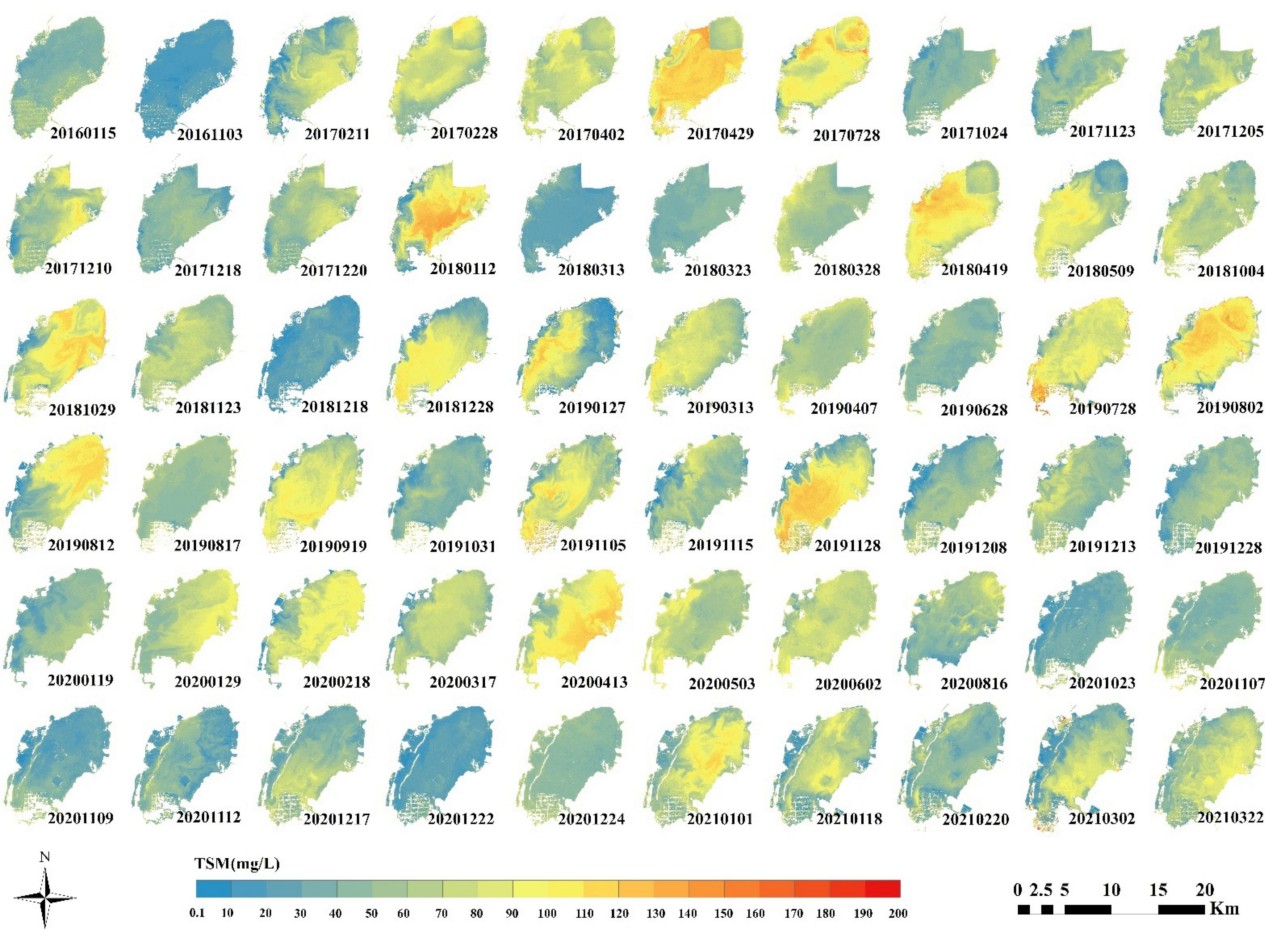

**Figure 7.** Spatial distribution of TSM concentration in Changdang Lake from 2016 to 2021.

### 3.3. Relationship between Ecological Dredging and Spatiotemporal Variations of TSM Concentration

Ecological dredging of Changdang Lake was commenced in 2017 to combat the threats posed by rapid urbanization and development of aquaculture. Sediment removal was predominantly aimed at the protected areas, estuaries, and lakeshore areas exhibiting considerable pollution. The process of ecological dredging included two stages. The first stage involved the ecological desilting of Changdang Lake Water Source Reserve, and the second stage was aimed at the elimination of pollution caused by rivers entering the lake on the west side of Changdang Lake. Dredging methods included the implementation of the pumping dry method and underwater construction in the first and second stages, respectively. Here, we combined the inversion results of TSM concentration in Changdang Lake recorded from 2017 to 2021 with monitoring results of the ecological dredging process to conduct a preliminary analysis of the effect of ecological desilting in Changdang Lake.

#### 3.3.1. First Stage of Ecological Dredging

In the first stage of ecological dredging, the northeast of Changdang Lake was considered as the desilting area. Considering that the intake of Changdang Lake Waterworks was located in the northeast of the lake, a construction cofferdam was built in the northeast corner of Changdang Lake to ensure water quality at the intake. Furthermore, the silt was removed after pumping was conducted to thoroughly eliminate heavy metal pollution from the sediment.

Figure 8 displays the spatial distribution of TSM concentration in the northeast lake area during the first stage of ecological dredging; Sentinel-2 images were used for monitoring purposes. On 11 February 2017, the cofferdam was under construction and the TSM



concentration was relatively high near the cofferdam area. On 28 February 2017, the spatial distribution of TSM concentration changed between the northeast corner of Changdang Lake and other areas under the influence of the cofferdam. This spatial difference was fully reflected on 29 April 2017, indicating that the constructed cofferdam could separate Changdang Lake into two distinct areas. On 28 July 2017, the northeast lake area was reported to present a high concentration of TSM, which might have been caused by the activity of construction workers pumping water into the area, resulting in lower water levels and continuous agitation; thus, high TSM concentrations were recorded. On 24 October 2017, the northeast lake area was transformed into a water-free area and pumping was ceased. From 23 November 2017 to 28 March 2018, workers continued to remove the silt from the water-free area. By 19 April 2018, the first stage of the silt removal project was completed, and workers conducted reinjection of water. The high TSM concentration recorded in this period might be related to the low amount of water injected, causing greater vulnerability to natural and anthropogenic influences. On 9 May 2018, after the completion of dredging and after successful refilling of the water, the TSM concentration in the northeast lake area was estimated and was found to be low, indicating good water quality. Therefore, ecological dredging might be considered to promote the improvement of Changdang Lake water quality to a certain extent.

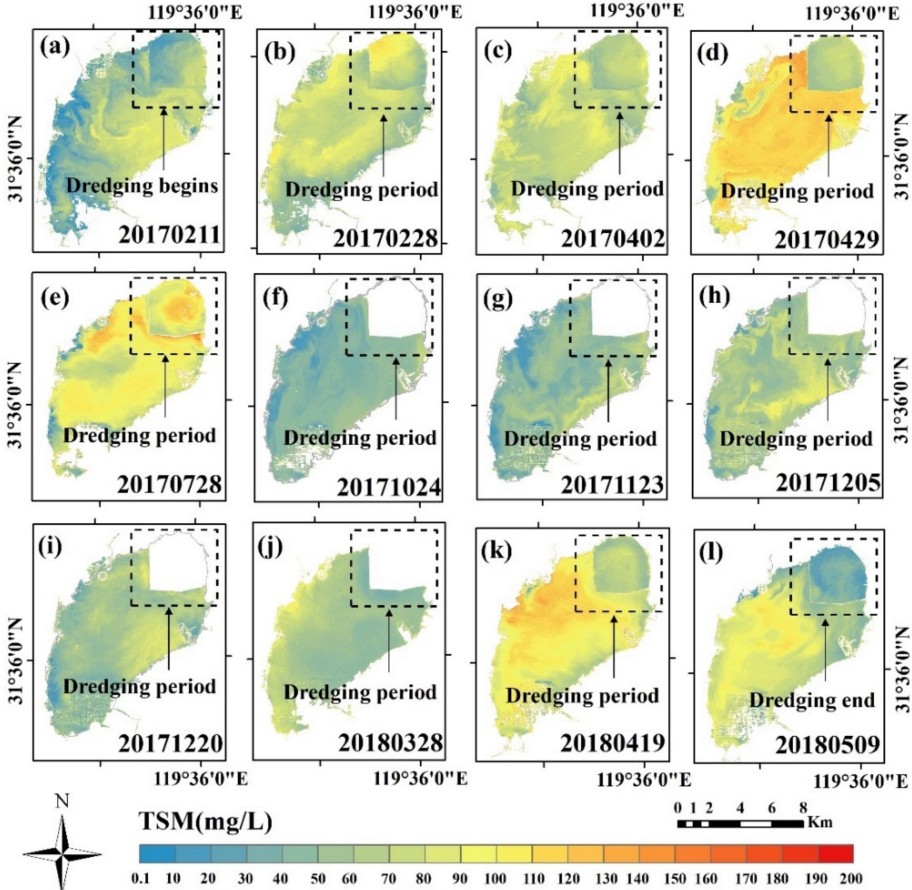

**Figure 8.** Sentinel-2 images showing temporal changes in the spatial distribution of TSM concentration in Changdang Lake during the first stage of ecological dredging. (**a**–**e**) indicates that construction cofferdams are being built in the northeast corner of Changdang Lake and may start pumping; (**f**–**j**) indicates the process of removing silt after pumping, and the white space indicates the period when there is no water; (**k**,**l**) indicates the process of re-injecting water into the northeast corner after silt removal is completed.

### 3.3.2. Second Stage of Ecological Dredging

The second stage of ecological dredging of Changdang Lake was conducted at the location where several rivers enter the lake on the west side of Changdang Lake. The stage was performed to remove considerably silted sediment from the mouth of the rivers, as it could markedly impact normal ship navigation and the quality of lake water, if not removed. The underwater dredging method was adopted [48].

Figure 9 displays the spatial distribution of TSM concentration in the western lake area during the second stage of ecological dredging; Sentinel-2 images were used to perform monitoring. As shown in Figure 9, no evident construction of the cofferdam was observed on 16 August 2020, as construction had not yet commenced. On 23 October 2020, an anhydrous cofferdam was observed in the western lake area. Furthermore, the TSM concentration near this area was relatively high, indicating that the cofferdam was under construction. From 7 November 2020 to 2 March 2021, the western lake area was subjected to a period of construction, which caused turbidity of the surrounding water body and increased the concentration of TSM. The second stage of the dredging project was continued in March 2021; therefore, the final effects of the second stage could not be determined. Change in water quality resulting from this dredging stage should be analyzed through conducting continuous follow-up monitoring and in-depth studies.

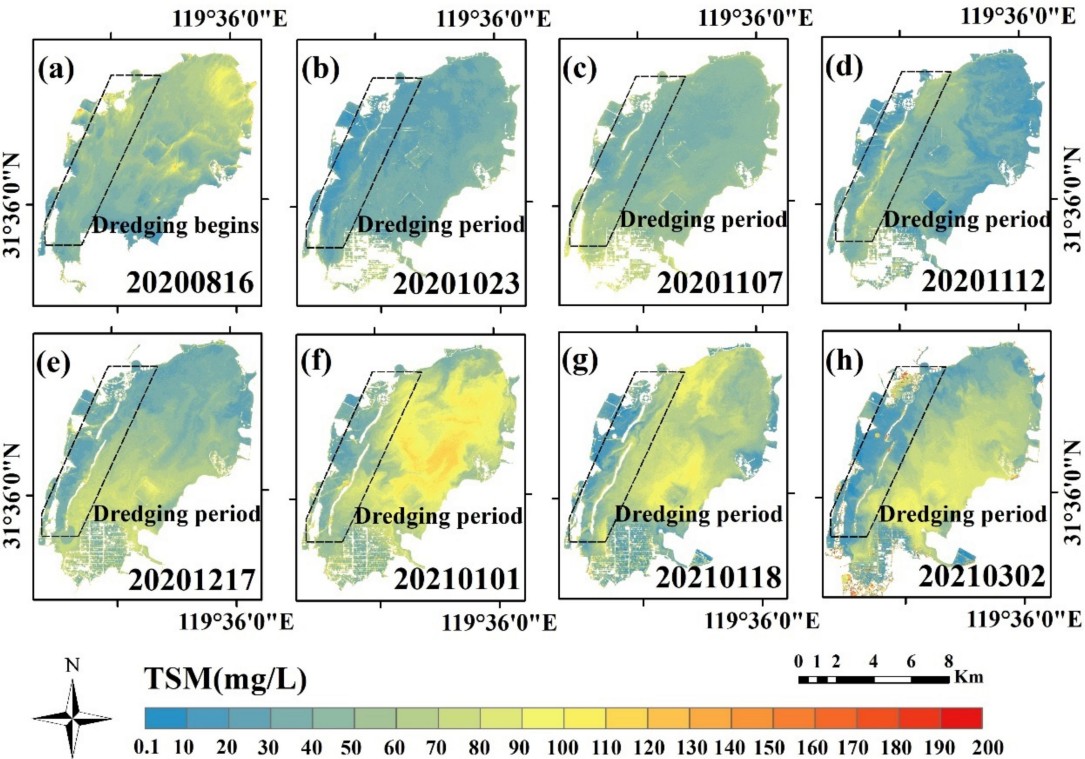

**Figure 9.** Sentinel-2 images showing temporal changes in the spatial distribution of TSM concentration in Changdang Lake during the second stage of ecological dredging. (**a**–**h**) indicates the process of building a cofferdam and removing silt in the western lake area of Changdang Lake.

## 4. Discussion

### 4.1. Comparison of TSM Concentration Models

Here, the inversion algorithms suitable for TSM concentrations in Changdang Lake were compared. Referring to the general TSM concentration inversion model proposed by other researchers in recent years, the measured data used in this study were incorporated into the corresponding model to estimate the TSM concentration in Changdang Lake.

Finally, after comparison with the accuracy of the Changdang Lake TSM concentration inversion model established in this paper, the calculated average relative error (MRE) and root mean square error (RMSE) are listed in Table 3. The input data comprise the remote sensing reflectance of Sentinel-2 images after atmospheric correction, where G = $R_{rs}$ (560 nm), R = $R_{rs}$ (665 nm), and NIR = $R_{rs}$ (842 nm).

**Table 3.** Comparison of published TSM concentration inversion models with the model proposed in this study.

| Reference | Model | MRE (%) | RMSE (mg/L) |
|-----------|-------|---------|-------------|
| Petus et al. [9] | $y = 2 \times 10^6 \times R^2 - 87{,}342 \times R + 972.04$ | 60.19 | 44.43 |
| Gernez et al. [10] | $y = 17.127 \times \exp[(\text{NIR}/\text{G}) \times 3.7221]$ | 95.27 | 59.09 |
| Shi et al. [12] | $y = 38.921 \times \exp(22.924 \times R)$ | 25.55 | 20.31 |
| Zheng et al. [34] | $y = 2448.6 \times \text{NIR} + 39.919$ | 37.20 | 22.43 |
| This study | $y = 184.14 \times (\text{NIR}/\text{R}) - 59.716$ | 20.49 | 14.96 |

According to the MRE and RMSE values, the TSM concentration inversion algorithm proposed in this study exhibited increased accuracy. Therefore, the proposed algorithm exhibited good applicability in the case of Changdang Lake and could provide an important reference for subsequent studies.

### 4.2. Seasonal Variations of TSM Concentration in Changdang Lake

The distribution of TSM concentration exhibited seasonal variations; therefore, the TSM concentration inversion model was used to retrieve effective data for each season in Changdang Lake. As fewer Sentinel-2 data were obtained in 2016 and 2021, it could not cover four seasons. Therefore, only the seasonal average TSM concentrations recorded from 2017 to 2020 were calculated. The seasonal average of TSM concentration is the average of TSM concentration retrieved from the available Sentinel-2 images for each season, from 2017 to 2020; that is, it has been averaged in time and space.

Figure 10 displays the changes recorded in the average and standard deviation of TSM concentration in Changdang Lake during the four seasons. The TSM concentration for the summer of 2018 was not calculated due to a lack of availability of good-quality Sentinel-2 images. Generally, TSM concentrations were higher in spring and summer, and were lower in autumn and winter. The estimated average and standard deviation of TSM concentrations in winter, spring, summer and autumn were 59.69 ± 15.52 mg/L, 69.58 ± 20.78 mg/L, 72.55 ± 18.44 mg/L and 57.43 ± 18.74 mg/L, respectively.

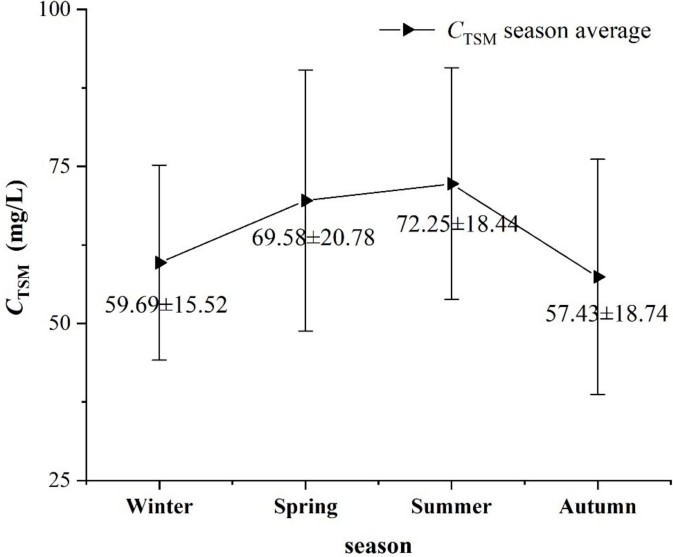

**Figure 10.** Average and standard deviation of TSM concentration in Changdang Lake in four seasons.

*4.3. Influence of Meteorological Factors on TSM Concentration in Changdang Lake*

Previous studies have shown that wind speed and precipitation can affect the temporal and spatial variations of TSM concentration in water bodies [12,49,50]. In this study, the daily average wind speed and daily precipitation data from 2017 to 2019 were obtained, in order to test whether the change in TSM concentration is related to wind speed and precipitation. Furthermore, the TSM concentration of 38 periods of Sentinel-2 images obtained from Changdang Lake from 2017 to 2019 was inversed, and the average TSM concentration of 38 periods of Sentinel-2 images obtained from the images. After six groups of outliers were removed, the correlation between the remaining 32 groups of data and TSM concentration was analyzed.

The correlation analysis exhibited a positive correlation between the average wind speed and TSM concentration ($R^2 = 0.5316$), as shown in Figure 11a. This may be related to the fact that Changdang Lake is a shallow lake vulnerable to the re-suspension of sediment due to the influence of wind speed, which makes the concentration of TSM increase in a short time. Wind speed might be the driving factor for the change in TSM concentration in Changdang Lake.

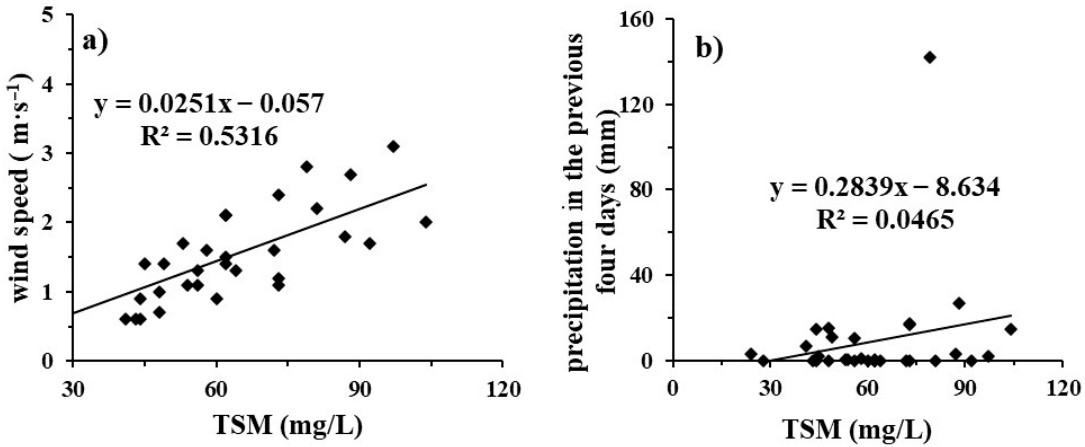

**Figure 11.** Correlation analysis between meteorological factors and TSM concentration. (**a**) Wind speed, and (**b**) precipitation recorded during the previous four days.

In this study, the correlation between the average TSM concentration of Changdang Lake retrieved from 32 periods of Sentinel-2 images from 2017 to 2019 and the total precipitation of the previous 4 days was analyzed. However, it was inferred that there is no obvious correlation between them, as shown in Figure 11b; that is, precipitation might not have a significant effect on the change in TSM concentration in Changdang Lake.

## 5. Conclusions

This study reports the use of in situ hyperspectral data and TSM concentrations of Changdang Lake to establish a TSM concentration inversion model based on the near-infrared to red band ratio and further apply it to the Sentinel-2 images of Changdang Lake. The temporal and spatial distributions of TSM concentrations were obtained from 2016 to 2021. The applicability of the model, the seasonal variability of TSM concentration (from 2017 to 2020), and the influence of meteorological factors were further analyzed. Furthermore, the effect of ecological dredging was preliminarily monitored by using remote sensing images based on the spatiotemporal variations of TSM concentration. The major conclusions of this study have been presented as follows.

(1) The accuracy of the TSM concentration inversion model based on ground-measured data and synchronous Sentinel-2 image data was high, with MRE and RMSE values of 15.14% and 10.07 mg/L, and 20.49% and 14.96 mg/L being obtained, respectively.

(2) The TSM concentration in Changdang Lake exhibited evident seasonal variations, exhibiting higher values in spring and summer and lower values in autumn and winter. By monitoring the ecological dredging process of Changdang Lake, it was found that the ecological dredging project might improve the water quality in Changdang Lake to some extent.

(3) The inversion results clearly revealed the temporal and spatial distributions of TSM concentration in Changdang Lake, confirming the application potential of Sentinel-2 images for the long-term monitoring of water quality changes in small- and medium-sized lakes.

**Author Contributions:** Conceptualization, Z.G. and Q.S.; Project Administration, Z.G. and X.W.; Writing—Original Draft Preparation, Z.G.; Writing—Review and Editing, Y.Y.; Validation, H.P. and M.W.; Supervision, D.S. and L.W.; Methodology, R.W., W.L. and J.S. All authors have read and agreed to the published version of the manuscript.

**Funding:** This research was supported by the National Natural Science Foundation of China (Grant Nos. 41871346) and Innovation and Entrepreneurship Program for college students of Jiangsu Ocean University, the titled is: "Study on the change of construction land of Nanjing in 30 years" (SD201911641110001).

**Institutional Review Board Statement:** Not applicable.

**Informed Consent Statement:** Not applicable.

**Data Availability Statement:** The data presented in this study are available on request from the corresponding author.

**Acknowledgments:** The authors would like to thank Qian Shen from the Key Laboratory of Digital Earth Science, Aerospace Information Research Institute, Chinese Academy of Sciences for the help with writing. We are also thankful to all anonymous reviewers for their constructive comments provided on the study.

**Conflicts of Interest:** The authors declare no conflict of interest.

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
