# Peer review of "Spatiotemporal Distribution of Total Suspended Matter Concentration in Changdang Lake Based on In Situ Hyperspectral Data and Sentinel-2 Images"

_remotesensing, doi:10.3390/rs13214230_

Round 1

Reviewer 1 Report

This manuscript presents a regional model for TSM estimation for Changdang Lake from Sentinel-2 image data. It then analyzes the spatio-temporal patterns in TSM  variability in terms of seasonality, meteorological conditions and  mechanical dredging.

While the data analysis that went into the manuscript is generally sound, there are a few areas within the text that can be improved with more detail. Particular comments are listed below. 

Section 1:

Although extensive references are made to retrieving TSM concentrations from image data acquired by various satellite sensors, there is no discussion on minimum sizes of water bodies that can adequately be analyzed from e.g. MODIS and Landsat sensors. The paragraph on P2L79-97, leads up to S2 image data as better suited for medium and small lakes but no mention is made of adjacency effects etc.

Section 2.1:

The annual average water level in the lake is stated to be 1.2-1.5m. How certain are the authors that analysis was done in optically deep water? i.e. where the bottom is not affecting the spectral response measured by the satellite? This would probably not be eliminated by SWIR correction and may affect the response in the red band where light penetrates deeper into the water column. 
Also how did the authors deal with the spectral effects of submerged aquatic vegetation on the signal (alluded to in P5L178-180)? In your pre-processing and masking steps, how did you ensure the elimination of adjacency effects of the surrounding terrestrial areas? 

Section 2.2.1

P5L147 (equation 1): How was Lsky calculated?

Section 2.3.2:

Equation 4: Is this a real linear relationship? Or does it appear linear because most of the samples have a moderate TSM concentration? There may be an increasingly under estimation of TSM in waters with high TSM concentration and in very clear waters, the TSM concentration may become over-estimated.  

Section 3.1

How was the spectra collected from the 03/05/2020 S2 image to do the validation for the image re-correction and the TSM retrieval accuracy?

Section 3.4

The average seasonal TSM concentration for each year was reported but not the spatial variability. A measure of the range and/or standard deviation added to Figure 9 would be useful in addition to the seasonal mean. 

Reviewer 2 Report

Dear Author,

The paper aims to develop models for TSM on basis of Sentinel-2 images integrated with in-situ measurement of spectral reflectance patterns of water using a spectroradiometer device. The paper is scientifically written and well-structured by illustrative figures and maps. A feasible flow was followed in presenting and discussing the results. Only few comments are requested as following;

  • Improve the title and I suggest adding the hyperspectral data in the title where the spectroradiometer data was integrated with Sentinel-2 data to develop the models.
  • Add a flow chart showing the methodology adopted in the paper.
  • In conclusion section, focus on the main findings of the study.

Reviewer 3 Report

This manuscript provides an analysis of total suspended matter (TSM) in a lake in China. Field-based observations in May 2020 provide a dataset of measured TSM as well as surface reflectance spectra for 20 reference locations. The paper also uses a time series of 60 Sentinel-2 images, acquired between 2016 and 2021. The authors provide a TSM retrieval model fitted on the basis of the above datasets, and provide an assessment of its accuracy, as well as a comparison of the results against other TSM algorithms from the existing literature. Finally, the paper presents an overview of the spatial and temporal characteristics of TSM over the lake and time window of interest.

This paper is well written, and provides an interesting insight into the mapping of water quality at large-scale from remote sensing imagery. For the most part, the methods used in the paper are technically sound and clearly explained. However, a few issues require clarification and/or improvement in the paper before it can be accepted for publication. These issues are listed in the following comments.

Line 256: I believe that the words “remote sensing” need to be removed here. As I understand, the comparison here is between the Sentinel-2 data and the *ground-based* surface reflectance data (which is not remotely sensed).

Methods: the ground-truth dataset is here very small. Splitting it further into two training/testing sets means that the results are highly susceptible to sampling bias (among others). For instance, the results presented in Fig. 4, with only 6 data points, are not very convincing (the results would likely be very different if 6 different data points had been randomly selected). I recommend that the authors use a k-fold cross-validation approach instead (e.g. k=5), which will provide a more robust estimate of accuracy with such a limited dataset. As a result, the proposed algorithm can also be fitted on the basis of the full dataset of 20 data points instead of only 16.

Section 3.1.4: it appears that the TSM retrieval formulae provided in Table 3 have been taken directly (i.e. without modification) from the corresponding references. As done in the current paper, these retrieval algorithms have likely been fine-tuned (i.e. fitted) on the basis of some very specific ground-truth data measured in the environments corresponding to the case studies carried out in those references. As such, it is completely to be expected that the exact same formulae will not perform well in the context of the current paper and its experimental setup, which likely represents a completely different environmental setting compared to the references in Table 3. What should be done instead is to fit these formulae again on the basis of the dataset considered in the present work (i.e. the 20 data points). For instance, the first formula in Table 3 essentially represents the generic model: a * R * R + b * R + c. The authors should thus fit the parameters a, b and c using their own ground-based data, and only then compare the resulting accuracy of this new model to the accuracy of the model proposed in the current paper (and this should obviously be repeated for the other 3 models presented in Table 3). This will ensure a fair comparison of results, and will provide a better technical basis for the authors’ claims and arguments.

Figures 9 and 10 (and related text): the paper is not always clear with regards to how the various datasets were aggregated and/or averaged. In Fig. 9 for instance, I assume that the TSM images from each season have been averaged both in space and time to achieve the points plotted in the graph. In other words, all the TSM pixels over the lake were used to calculate the average for a given season. While this is relatively straightforward, this should be clearly explained in the text. Also, in Fig. 10: where do the TSM values come from? As only 35 odd points are displayed, I can only assume that these correspond to the available Sentinel-2 images between 2017 and 2020, and that each TSM map has been averaged spatially to yield one averaged TSM value per date. Please update the text to explain this process more clearly.

A common error made when analysing the type of results provided in this paper is to assume that correlation means causation (which of course isn’t the case). On lines 408-409, as well as 439, the authors declare that wind speed was shown in the paper to be a key driver (influencing factor) of TSM, which is incorrect: all the authors have done is to show a correlation between wind speed and TSM, and this does not prove that an increase in wind speed leads to increased TSM measurements (proving causation is usually a lot harder than simply demonstrating correlation). These statements should thus be corrected to avoid the false claims that this paper has demonstrated causation. (And note that on the other hand, it is OK to *speculate* that wind speed *might* be the driver of TSM).

Line 414: I do not see how the authors observe a “good correlation between TSM and precipitation”… In a linear model with only one predictor, the correlation coefficient is the square root of the coefficient of determination, so here, the correlation is roughly 0.2, which is not, in my view, an indication of “good correlation”.
